# Nonlinear responses of particulate nitrate to $NO_x$ emission controls in the megalopolises of China

Mengmeng Li[1, *], Zihan Zhang[1], Quan Yao[2], Tijian Wang[1], Min Xie[1], Shu Li[1],

Bingliang Zhuang[1] and Yong Han[3]

[1] School of Atmospheric Sciences, Nanjing University, Nanjing 210023, China

[2] Statistical Bureau for the Qingjiangpu District, Huaian 223001, China

[3] Guangdong Province Key Laboratory for Climate Change and Natural Disaster Studies, School of Atmospheric Sciences, Sun Yat-Sen University, Guangzhou 510000, China

[*] Corresponding author: mengmengli2015@nju.edu.cn

**Abstract**

Nitrate is an increasingly important component of fine particulate matter ($PM_{2.5}$) in Chinese cities. The production of nitrate is not only related to the abundance of its precursor but also supported by the atmospheric photochemical oxidants, raising a new challenge to the current emission control actions in China. This paper uses comprehensive measurements and a regional meteorology-chemistry model with optimized mechanisms to establish the nonlinear responses between particulate nitrate and nitrogen oxides ($NO_x$) emission controls in the megalopolises of China. Nitrate is an essential component of $PM_{2.5}$ in eastern China, accounting for 9.4–15.5% and 11.5–32.1% of the $PM_{2.5}$ mass for the warm and cold seasons. The hypothetical $NO_x$ emission reduction scenarios ($-10\%\sim-80\%$) during summer-autumn result in almost linearly lower $PM_{2.5}$ by $-2.2\%$ in Beijing-Tianjin-Hebei (BTH) and $-2.9\%$ in Yangtze River Delta (YRD) per 10% cut of $NO_x$ emissions, whereas they lead to a rather complicated response of PM components in winter. Wintertime nitrate is found to increase by $+4.1\%$ in BTH and $+5.1\%$ in YRD per 10% cut of $NO_x$ emissions, with

nearly unchanged nitric acid ($HNO_3$) and higher dinitrogen pentoxide ($N_2O_5$)
intermediate products produced from the increased atmospheric oxidants levels. An
inflexion point appears at 40–50% $NO_x$ emission reduction, and a further cut in $NO_x$
emissions is predicted to cause −10.5% reduction of nitrate for BTH and −7.7% for
YRD per 10% cut of $NO_x$ emissions. In addition, the 2012–2016 $NO_x$ control strategy
actually leads to no changes or even increases of nitrate in some areas (8.8% in BTH
and 14.4% in YRD) during winter. Our results also emphasize that ammonia ($NH_3$)
and volatile organic compounds ($VOC_s$) are effective in controlling nitrate pollution,
whereas decreasing the sulfur dioxide ($SO_2$) and $NO_x$ emissions may have counter-
intuitive effects on nitrate aerosols. This paper helps understand the nonlinear aerosol
and photochemistry feedbacks, and defines the effectiveness of proposed mitigations
for the increasingly serious nitrate pollution in China.

## 1 Introduction

Secondary inorganic aerosols (SIA), including sulfate ($SO_4^{2-}$), nitrate ($NO_3^-$) and ammonium ($NH_4^+$) account for 30–60% of the total fine particulate matter ($PM_{2.5}$) mass during haze events in China (Huang et al., 2014a; Zhao et al., 2013). Since the enactment of the Air Pollution Action Plan in 2013, the Chinese government has taken drastic measures to reduce the emissions of sulfur dioxide ($SO_2$), nitrogen oxides ($NO_x$) and primary $PM_{2.5}$, leading to significant decreases in sulfate and overall $PM_{2.5}$ concentrations in cities (Silver et al., 2018; Li et al., 2021a; Wang et al., 2017b). Meanwhile, the nitrogen/sulfur (N/S) ratio in $PM_{2.5}$ increased significantly and nitrate had been the main component of $PM_{2.5}$ (16–45%) during haze episodes, despite a more than 20% reduction in the concentrations of its precursor $NO_x$ (Shao et al., 2018; Wen et al., 2018; Zhai et al., 2019). The increasingly serious nitrate pollution has emerged to be the new emphasis of air pollution controls in China.

Nitrate formation involves complex multiphase chemical reactions. In the daytime, nitrogen dioxide ($NO_2$) reacts with hydroxyl radical (OH) to produce nitric acid ($HNO_3$). With excess ammonium ($NH_3$), low temperature and insufficient sulphuric acid, this reaction can proceed quickly and produce high ammonium nitrate (Seinfeld and Pandis, 2006). In the nighttime, however, high-concentration $NO_2$ reacts with ozone ($O_3$) to produce the nitrate radical ($NO_3$) and dinitrogen pentoxide ($N_2O_5$). The heterogeneous hydrolysis of $N_2O_5$ on wet particles is the main pathway for nocturnal nitrate formation (56–97%) (He et al., 2018; Pathak et al., 2011; Xue et al., 2014).

Nitrate chemistry is not only related to the abundance of its precursor $NO_x$, but also supported by the atmospheric oxidants (e.g., OH and $O_3$) produced from the photochemical reactions of $NO_x$ and volatile organic compounds ($VOC_s$) (Meng et al.,

1997). Using a box model, some studies have determined that the relationship
between particulate nitrate and $NO_x$ emissions is nonlinear depending on the ozone
chemical sensitivity regime (Pun and Seigneur, 2001; Nguyen and Dabdub, 2002).
Pun and Seigneur (2001) showed that the daytime $HNO_3$ production was more
sensitive to the concentrations of atmospheric oxidants, and that in the VOC-limited
regime the decrease of $HNO_3$ production due to the $NO_x$ emission control might be
offset by the increase of OH. Nguyen and Dabdub (2002) calculated the detailed
isopleth between nitrate and $NO_x$ emissions; they found that the reduction of $NO_x$
emissions resulted in a decrease of nitrate in the $NO_x$-limited regime, and an increase
of nitrate under extreme conditions in the VOC-limited regime. Despite that, the
single-site box model results could not distinguish the regional differences among
chemical regimes; the basic hypotheses in box models to predict nitrate production are
also unreasonable in the real atmosphere.

As an important precursor for both fine particles and ozone, the strict control of

$NO_x$ emissions has started in China since the 12[th] Five-Year Plan (Zheng et al., 2018).
A confounding factor is that, for most cities in China, the production of $O_3$ is usually
limited by $VOC_s$ (Xie et al., 2014; Dong et al., 2014; Liu et al., 2010). The control of
$NO_x$ emissions has therefore resulted in an increase of surface $O_3$ concentrations in
recent years (Li et al., 2021a; Li et al., 2019a; Kalsoom et al., 2021), implying
complex impacts on nitrate formation. Li et al. (2021a) and Liu and Wang (2020)
examined the influencing factors on the surface $O_3$ trends in China from 2013 to 2017
using regional chemical models. They highlighted that the control of $NO_x$ emissions
explained 11–35% of the increased $O_3$ due to the nonlinear $NO_x$-$VOC_s$-$O_3$ chemistry,
and that for most regions the magnitudes could be comparable to those resulting from
the meteorological influences and aerosol effects. Some simulations thought that the
$NO_x$ emission increase in 2005–2012 resulted in an increase of nitrate by 3.4% $yr^{-1}$ in
eastern China (Geng et al., 2017; Wang et al., 2013), and the following $NO_x$ emission
control resulted in a decrease of nitrate by 3–14% (Wang et al., 2014). Recent
evidence from field observations (Fu et al., 2020) and numerical simulations (Dong et
al., 2014), however, suggested that the $NO_x$ emission reduction in China could result
in an increase of nitrate in winter through increased photochemical oxidants and
nocturnal $N_2O_5$ chemistry, but a decrease in other seasons. In the next 5–10 years,
$SO_2$ emissions might level off in China, while $NO_x$ emissions will become stringently
controlled to ensure further air quality improvements (Zheng et al., 2018). Accurately
understanding the nonlinear aerosol and photochemistry feedbacks is crucial to
resolve the emerging nitrate pollution and to establish reasonable air pollution control
strategies in China.

To address this issue, we use comprehensive measurements and a regional

meteorology-chemistry model combined with hypothetical $NO_x$ emission scenarios to
establish the nonlinear response relationships between particulate nitrate and $NO_x$
emission controls in the megalopolises of China. The model configurations, numerical
designs and observational data are presented in Sect. 2. Sect. 3 discusses the results.
Finally, a summary is presented in Sect. 4.
**2 Materials and Methods**
**2.1 Model setup and experimental designs**
This study uses the Weather Research and Forecasting-Chemistry (WRF-Chem)
model version 4.1 developed by Grell et al. (2002) to simulate the regional
meteorology and atmospheric chemistry. The mesoscale meteorology and air quality
simulations of WRF-Chem have been improved in terms of incorporating the satellite-
derived land surface parameters (Li et al., 2014; Li et al., 2017), and optimizing the
SIA formation pathways enhanced by mineral aerosols (Li et al., 2019b; Huang et al.,
2014b).

The modeling domain covers two main megalopolises of China and its adjacent

areas—the Beijing-Tianjin-Hebei (BTH) region and the Yangtze River Delta (YRD)
region (Fig. 1). The modeling framework is configured with 81×86 grid cells at 25 km
horizontal resolution. The model is run with an 84-hour model cycle, with the first 12
hours discarded as spin-up time and model outputs of each model cycle to provide
chemical initial conditions for the subsequent overlapping 84-hour simulation. The 6-
hour, $1°×1°$ National Centers for Environmental Prediction Final (NCEP/FNL)
analysis fields are regularly input for the model initial and lateral boundary
meteorological conditions.

The model physical configurations include the YSU boundary layer scheme

(Noh et al., 2003), the RRTMG radiation scheme (Iacono et al., 2008), the Noah land
surface scheme (Ek et al., 2003) and the Lin microphysics scheme (Lin et al., 1983).
We have updated the land cover type and vegetation data in WRF mesoscale model
with the latest land surface parameters derived from Moderate Resolution Imaging
Spectroradiometer (Li et al., 2014; Li et al., 2017).

The atmospheric chemistry is simulated using the Carbon Bond Mechanism

version Z (CBMZ) (Zaveri and Peters, 1999) gas-phase chemistry module coupled
with a four-bin sectional Model for Simulating Aerosol Interactions and Chemistry
(MOSAIC) (Zaveri et al., 2008). The aqueous-phase chemistry is based on the
Carnegie Mellon University (CMU) scheme including 50 species and more than 100
reactions (Fahey and Pandis, 2001). Formation of SIA in the default WRF-Chem
model accounts for the gas-phases oxidation of $SO_2$ and $NO_2$, and aqueous-phase
oxidation of $SO_2$ by hydrogen peroxide ($H_2O_2$) and $O_3$ in cloud. We have optimized
the SIA formation pathways by including the aqueous $SO_2$ oxidation catalyzed by
mineral ions and heterogeneous uptakes of $SO_2$, $NO_2$, $NO_3$, $N_2O_5$ and $HNO_3$ on
mineral aerosols in the MOSAIC aerosol module (Li et al., 2019b; Huang et al.,
2014b).
Anthropogenic emissions are adopted from the 2016 Multi-resolution Emission
Inventory for China (MEIC) and the 2010 MIX-Asia emission inventory for regions
outside of mainland China developed by Tsinghua University (http://meicmodel.org).
Biogenic emissions are calculated online using the Model of Emissions of Gases and
Aerosols from Nature (Guenther et al., 2006).
A series of WRF-Chem simulations is designed as summarized in Table 1. In the
baseline simulation (denoted as the B0 scenario), the anthropogenic emissions in
China remain unchanged at the usual levels in 2016. Simulation N0 is the same as B0,
but it only considers the gas-phase oxidation production of $HNO_3$ ($NO_2+OH \rightarrow HNO_3$)
and its subsequent partitioning to the aerosol phase of nitrate in WRF-Chem. The B0
and N0 simulations are combined to distinguish the contributions of gas-phase
oxidation and heterogeneous pathways (i.e., uptakes of $N_2O_5$, $NO_3$ and $NO_2$) for the
formation of nitrate aerosols during the warm and cold seasons. A group of sensitivity
scenarios (C1~C8) are designed with the perturbed anthropogenic $NO_x$ emissions in
China cut by 10%, 20%...and 80%, respectively. The differences between B0 and
C1~C8 simulations are calculated to illustrate the responses of particulate pollution in
China's megacities to the $NO_x$ emission reduction scenarios. Another simulation (E1)
is designed with the anthropogenic emissions of $NO_x$ in China set to the 2012 levels
to show the impacts of 2012–2016 $NO_x$ control strategy on particulate pollution.
Additionally, in order to evaluate the effectiveness of multi-pollutants cooperative
controls, three series of simulations ($C_{S-N}$, $C_{N-N}$ and $C_{V-N}$) are also supplemented with
the anthropogenic emissions of $SO_2$, $NH_3$ and $VOC_s$ in China cut by 20%, 40%...and
80%, respectively. The differences between B0 and $C_N$/$C_{S-N}$/$C_{N-N}$/$C_{V-N}$ simulations are
calculated to illustrate the responses of nitrate pollution in China's megacities to the
multi-pollutants cooperative controls.

For all simulation scenarios, two month-long periods during the Campaign on

Air Pollution and Urban Meteorology in Yangtze River Delta (CAPUM-YRD)—
August 15 to September 16 (Period I) and November 24 to December 26 (Period II) in
2016, are simulated to represent the warm and cold seasons, respectively (Shu et al.,
2019). The complete simulation consists of thirteen 84-hour model cycles with the
first 6 days as spin-up for chemistry and the remaining model outputs for analysis.
**2.2 Weather and air pollutants data**

Surface meteorological observations at 186 land-based automatic stations across

China (Fig. 1) are collected for model meteorological validation, including hourly
data of 2 m air temperature, 2 m relative humidity and 10 m wind speed. These data
are archived at the U. S. National Climatic Data Center (NCDC) (Smith et al., 2011).

Air pollutants data at the national air quality monitoring network and regional

supersites of China (Fig. 1) are used for model chemical validation. This nationwide
monitoring network contains 1597 sites covering 454 cities in mainland China, as
shown in Fig. 1. Six routine air pollutants including $PM_{2.5}$, particulate matter with
aerodynamic diameter less than 10 μm ($PM_{10}$), $SO_2$, $NO_2$, carbon monoxide (CO) and
$O_3$ are monitored and reported hourly by Chinese National Environmental Monitoring
Center (CNEMC) network (available at http://websearch.mep.gov.cn/).

Additionally, four comprehensive atmospheric environment supersites in YRD

including Dianshanhu (DSH; 31.1°N, 121.0°E), Pudong (PD; 31.2° N, 121.5°E),
Nanjing (NJ; 32.1°N, 118.8°E) and Hangzhou (HZ; 30.3°N, 120.2°E) measured the
mass concentrations of $PM_{2.5}$, water-soluble ions (sulfate, nitrate, ammonium, sodium,
chloride, potassium, calcium and magnesium), carbonaceous aerosols (elemental
carbon (EC) and organic carbon (OC)) and gaseous pollutants ($SO_2$, $NO_2$, CO and $O_3$)
during the CAPUM-YRD campaign. Details for the methods and data at the four
supersites are described in Shu et al. (2019).

**3 Results and discussions**

**3.1 Model weather and chemical validation**

Model evaluations indicate that the WRF-Chem model is able to simulate the
weather and atmospheric pollution characteristics in China. The simulated magnitudes
of surface temperature by WRF-Chem in general agree with actual observations, with
a correlation efficient ($R$) of 0.89 and 0.94, and a normalized mean bias (NMB) of
−0.55% and −0.80% respectively in Period I and Period II (Table 2). Underestimation
of relative humidity (−5.65% in Period I and −6.56% in Period II) is common in the
WRF simulation and it might be attributed to the influence of the boundary layer
parameterization on the weather forecast (Bhati and Mohan, 2018; Gomez-Navarro et
al., 2015). Clear overestimation of wind speed (23.72% in Period I and 40.64% in
Period II) might be because of the unresolved topography in WRF (Jimenez et al.,
2013; Li et al., 2014).
The predicted concentrations of routine air pollutants also faithfully capture the
spatial and seasonal patterns of observed surface $PM_{2.5}$, $SO_2$, $NO_2$ and $O_3$ levels in
both seasons (Fig. 2). Both simulations and observations display high air pollutants
concentrations in the vicinity of North China Plain (NCP) and eastern China, but with
higher $O_3$ levels in the warm season and oppositely higher $PM_{2.5}$ and other gaseous
pollutants concentrations in winter. The model statistical evaluations show a mean
bias (MB) of −3.66, −1.14, 4.70 and 18.32 μg m$^{-3}$, and NMB of −9.92, −6.46, 16.47
and 7.72% for $PM_{2.5}$, $SO_2$, $NO_2$ and $O_3$ in Period I, and a relatively larger MB of
−27.31, −11.65, 1.27 and −39.01 μg m$^{-3}$, and NMB of −29.82, −28.11, 2.40 and
−31.05% in Period II, respectively (Table 3). The uncertainty in emissions data, the
absence of secondary organic aerosol in MOSAIC aerosol chemistry or the simulated
wind errors (Table 2) may be responsible for the larger atmospheric chemical biases
in winter, which has been extensively discussed in some studies (Zhao et al., 2016; Li
et al., 2021a).
As the most important components of $PM_{2.5}$, reasonable representation of SIA is
imperative to $PM_{2.5}$ simulation. Evaluations with measurements of $PM_{2.5}$ components
at the four supersites of eastern China show that the model performs reasonably in
simulating the seasonal variations and proportions of aerosol species in $PM_{2.5}$, but it is
biased low by 10–40% in simulating the magnitudes of SIA concentrations (Fig. 3).
The model underestimation is −1.8, −2.2 and −2.2 μg m$^{-3}$ for sulfate, nitrate and
ammonium, respectively, in Period I, and −2.6, −4.3 and −3.4 μg m$^{-3}$ in Period II. The
model also captures the large change of N/S ratio from the warm to cold seasons, that
increases from 0.4 in Period I to 1.6 in Period II. Our previous work (Li et al., 2019)
has confirmed that the consideration of the optimized aqueous and heterogeneous SIA
formation pathways in WRF-Chem significantly reduces the model biases by 41.4%
for sulfate and 44.6% for nitrate during the CAPUM-YRD campaign of 2016. Recent
studies highlighted that the remaining SIA simulation biases may be attributed to the
missing aqueous oxidation of $SO_2$ by $NO_2$ on alkaline aerosols under humid
conditions (Wang et al., 2016; Cheng et al., 2016).
**3.2 Air pollution and aerosol composition characteristics**
Chemical composition analyses of major gaseous and particulate air pollutants
suggest large seasonal variations of air pollution characteristics in China (Fig. 2).
Mainly emitted from combustion sources, atmospheric pollutants accumulate in the
densely industrialized and populated megalopolises of China, with a hotspot along
Beijing, Hebei, Shandong and their adjacent cities frequently exceeding China's
National Ambient Air Quality Standards. The average concentrations of surface $PM_{2.5}$,
$SO_2$, $NO_2$ and daily-maximum $O_3$ in China's routine air quality monitoring network
are 33.8, 15.8, 26.5 and 223.2 $\mu g\ m^{-3}$ for Period I, and 80.2, 34.7, 47.7 and 131.4 $\mu g$
$m^{-3}$ for Period II. The surface $PM_{2.5}$, $SO_2$ and $NO_2$ concentrations show obvious
increases by 137.6%, 119.2% and 80.2% during winter compared to those of the
summer-autumn period (Period I). The maximum surface $PM_{2.5}$ concentrations
recorded in the winter period was more than 600 $\mu g\ m^{-3}$, which is the highest value
ever recorded in 2016 and leads to the "orange" air quality alert.
The further analyses of $PM_{2.5}$ mass concentrations, major $PM_{2.5}$ components and
gases at the four supersites in YRD are presented in Fig. 4–5. Organic matter (OM) is
obtained by multiplying the OC concentrations by a factor of 1.6, mainly accounting
for the hydrogen and oxygen masses in OM. The measured SIA concentrations
exhibit high levels, with average values of 18.8 $\mu g\ m^{-3}$ for Period I and 37.1 $\mu g\ m^{-3}$
for Period II. The three SIA components together account for 32.3–57.4% (48.6% on
average) and 27.7–70.9% (56.9% on average) of the total $PM_{2.5}$ mass concentrations,
and become the primary components of $PM_{2.5}$ in the two periods. The proportions of
sulfate, nitrate and ammonium in total $PM_{2.5}$ range from 13.5–28.9%, 9.4–15.5% and
9.4–14.9% at the four supersites for Period I, and 9.2–20.3%, 11.5–32.1% and 7.0–
19.8% for Period II, respectively. The strikingly higher proportion of nitrate than that
of sulfate in $PM_{2.5}$ during winter, with a N/S ratio of 1.6, is in accordance with recent
observations during other winter haze periods in China (Shao et al., 2018; Zhang et al.,
2018; Zhang et al., 2019). They emphasized that since the enactment of Clean Air
Action Plan in 2013, the $PM_{2.5}$ components had changed clearly with decreasing
contributions from coal combustion.

The high proportions of sulfate and nitrate in $PM_{2.5}$ could be related to the high

oxidation rates of $SO_2$ and $NO_2$. The observed average values of sulfur oxidation ratio
(SOR=$[SO_4^{2-}]/([SO_4^{2-}]+[SO_2])$) and nitrogen oxidation ratio (NOR=$[NO_3^-]/([NO_3^-$
$]+[NO_2])$) are 0.41 and 0.13 in Period I, and 0.33 and 0.21 in Period II. In contrast, the
observed SOR is generally higher in summer-autumn than winter, opposite to that of
NOR, indicating the enhanced formation of nitrate in winter. Shu et al. (2019) also
noted similar seasonal distinctions for SOR and NOR in YRD. They attributed the
weakened conversion from $NO_2$ to nitrate in summer to the volatility and evaporative
loss of nitrate (Sun et al., 2012). The sharp increase of particles and moderate ambient
humidity in winter also benefit the heterogeneous formation of SIA, leading to high
NOR and SOR (Wang et al., 2012).

Figure 6 illustrates the contributions of gas-phase oxidation and heterogeneous

reactions for the nitrate production calculated from B0 and E0 simulations. It is shown
that on a daily basis the gas-phase oxidation production of $HNO_3$ and its subsequent
partitioning to the aerosol phase is the principal formation route for particulate nitrate,
with the average contributions of 60.2% for BTH and 91.7% for YRD in Period I and
75.1% for BTH and 85.9% for YRD in Period II. The heterogeneous hydrolyses of
$N_2O_5$ and other nitrogenous gases (calculated as the model differences between B0
and N0 simulations) contribute to the remaining nitrate, particularly in BTH with high
aerosol loading. These calculated results (60.2–91.7% for $NO_2$+OH oxidation and
8.3–39.8% for heterogeneous pathways) are in line with previous assessments in
China and globally. Alexander et al. (2009) reported that the global tropospheric
nitrate burden is dominated by $NO_2$+OH (76%), followed by $N_2O_5$ hydrolysis (18%);
but recent results suggested that $N_2O_5$ hydrolysis was as important as $NO_2 + OH$ (both
41 %) for global nitrate production (Alexander et al., 2020). In major Chinese cities, it
was estimated that the conversion of $NO_x$ to nitrate was dominated by $NO_2 + OH$
oxidation in Shanghai, with a mean contribution of 55–77% in total and even higher
(84–92%) in summer (He et al., 2020). In NCP, the nitrate contribution of
heterogeneous pathways was about 30.8% (Liu et al., 2020) or even comparable to the
partitioning of $HNO_3$ (Wang et al., 2019; Wang et al., 2017a; Luo et al., 2021). The
nitrate formation from heterogeneous pathways is moderately underestimated in the
optimized WRF-Chem model of this study, possibly due to the uncertainties of
heterogeneous uptake coefficients and unclear reaction mechanisms applied in the
model (Li et al., 2019b; Xue et al., 2016; He et al., 2014).
**3.3 Nonlinear responses of nitrate to $NO_x$ emissions and their policy implications**
**3.3.1 $PM_{2.5}$-$NO_x$ and $O_3$-$NO_x$ responses in the warm and cold seasons**
$NO_x$ is key in atmospheric chemistry and serves as an important precursor for
both ozone and secondary aerosols. We conduct a series of simulations (C1~C8) with
perturbed $NO_x$ emissions to assess the responses of $PM_{2.5}$ mass concentrations to $NO_x$
emissions in two megalopolises of China (Fig. 7). The WRF-Chem simulation results
show that the responses of surface $PM_{2.5}$ concentrations to $NO_x$ emissions vary in
different seasons and display strong nonlinear behaviour in winter. To better quantify
their effectiveness, we define the $NO_x$ emission control efficiency (β), which denotes
the percentage changes of surface $PM_{2.5}$ or its components concentrations in response
to the successive 10% cut of $NO_x$ emissions.
In Period I (Aug–Sep), the $PM_{2.5}$-$NO_x$ responses are closer to a linear function,
reflecting a stronger sensitivity to the $NO_x$ emission changes in the warm season. The
surface $PM_{2.5}$ concentrations decrease almost linearly as we gradually reduce $NO_x$
emissions in China, with the average β values of −2.2% in BTH and −2.9% in YRD.
However, the PM$_{2.5}$-NO$_x$ emission responses in Period II (Nov–Dec) display strong
nonlinearity and are analogous to a quadratic parabola distribution for both regions.
The NO$_x$ emission reductions within the first 50% would even increase surface PM$_{2.5}$
concentrations by +1.2% averagely in BTH, and this β value increases to +1.8% in
YRD with the first 40% reductions of NO$_x$ emissions. Subsequently, the PM$_{2.5}$
responses shift towards a similar linear pattern, with an average β value of −2.5% in
BTH and −4.0% in YRD.

The distinct forms of PM$_{2.5}$-NO$_x$ emission responses for the warm and cold

seasons are determined by the seasonal ozone chemical sensitivity regimes. The
photochemical indicator of $\Delta[O_3]_{NOx}/\Delta[O_3]_{VOCs}$ with a critical value of 1.0 is used to
investigate the season-varying ozone sensitivity in China, which is calculated as the
ratio of ozone concentration changes under 20% NO$_x$ emission reduction to that under
20% VOC$_s$ emission reduction (Fig. S1). The results indicate a strong VOC-limited
ozone chemistry across China during winter, while either VOC-limited regime over a
large portion of NCP and eastern China or NO$_x$-limited regime in northern and
western China during summer-autumn, as also indicated from previous studies (Xie et
al., 2014; Dong et al., 2014; Liu et al., 2010). We find larger O$_3$ and OH productions
under the NO$_x$ emission reduction conditions in both seasons (Fig. 8–9), particularly
in Period II (Nov–Dec) with an average increase rate of +14.7% and +18.5% in BTH
and +25.2% and +23.1% in YRD per 10% cut of NO$_x$ emissions. The SIA formation
chemistry is highly limited by atmospheric oxidants produced from the NO$_x$-VOC$_s$-O$_3$
photochemical cycles. The nonlinear O$_3$-NO$_x$ responses indicate a rather complicated
aerosol and photochemistry feedback in megacities.
**3.3.2 Nonlinear responses of particulate nitrate to NOx emissions**
The SIA formation is basically driven by the atmospheric oxidants levels, and a
reduction of $NO_x$ emissions may have counter-intuitive effects on SIA components by
controlling the atmospheric oxidants levels. The calculated SIA components for each
emission scenario in both months show that the surface nitrate aerosols can be
substantially decreased/increased with reducing $NO_x$ emissions, but the sulfate and
ammonium concentrations have moderately smaller changes (Fig. 8–10).
Response of sulfate to the $NO_x$ emissions is more predictable and determined by
the changes of atmospheric oxidants levels since that the conversion of $SO_2$ to sulfate
is partly driven by OH in the gas-phase and by dissolved $H_2O_2$ or $O_3$ in the presence
of fog or cloud. In Period I (Aug–Sep), the sulfate-$NO_x$ response follows a gradual
quadratic parabola distribution as that of $O_3$-$NO_x$ and OH-$NO_x$ response curves (Fig.
8 and Fig. 10), with a fitted function in Eq. 1. The $\beta$ values for surface sulfate change
by −0.7%~+1.2% in BTH and −1.5%~+0.2% in YRD under each $NO_x$ emission
reduction scenarios.
$$[SO_4^{2-}] = -2.4\Delta E_{NOx}^2 - 1.7\Delta E_{NOx} + 6.1 \text{ in BTH} \quad (R^2=0.93) \qquad \text{(Eq. 1)}$$
$$[SO_4^{2-}] = -2.3\Delta E_{NOx}^2 - 0.9\Delta E_{NOx} + 6.8 \text{ in YRD} \quad (R^2=0.99)$$
where $[SO_4^{2-}]$ is the surface mean concentration of sulfate ($\mu g\ m^{-3}$); $\Delta E_{NOx}$ is the
percentage change of $NO_x$ emissions (%).
As expected, the production of nitrate reflects a strong sensitivity to $NO_x$ and it
decreases linearly with the $NO_x$ emission control, with an average $\beta$ value of −10.2%
in BTH and −11.5% in YRD, which further leads to a decrease of ammonium
concentrations by −3.3% in BTH and −4.3% in YRD (Fig. 8 and Fig. 10). The
formation of nitrate mainly involves the $NO_2$+OH→$HNO_3$ gas-phase oxidation and
the heterogeneous hydrolysis of $N_2O_5$ and other nitrogenous gases. The strong
sensibility of particulate nitrate in response to the $NO_x$ emission decreases can be
explained by the synchronously suppressive production of its intermediate products
$HNO_3$ and $N_2O_5$. For example, when the $NO_x$ emission is cut by 20%, the surface
$NO_2$ concentration in BTH drops by 20.0% but the surface $O_3$ and OH concentrations
increase slightly by 2.6% and 5.3% due to the reduction of $NO+O_3$ titration reaction
and the greater VOC availability in the warm season, leading to substantial reductions
in surface $HNO_3$ (−16.7%) and $N_2O_5$ (−8.9%) concentrations.
$$[NO_3^-]=-34.5\Delta E_{NOx}^2-23.8\Delta E_{NOx}+13.2 \text{ in BTH} \quad (R^2=0.84) \quad \text{(Eq. 2)}$$
$$[NO_3^-]=-36.5\Delta E_{NOx}^2-19.6\Delta E_{NOx}+12.0 \text{ in YRD} \quad (R^2=0.99)$$
$$[NH_4^+]=-9.1\Delta E_{NOx}^2-6.9\Delta E_{NOx}+6.2 \text{ in BTH} \quad (R^2=0.78) \quad \text{(Eq. 3)}$$
$$[NH_4^+]=-10.5\Delta E_{NOx}^2-6.2\Delta E_{NOx}+5.3 \text{ in YRD} \quad (R^2=0.98)$$
where $[NO_3^-]$ and $[NH_4^+]$ are the surface mean concentrations ($\mu g\ m^{-3}$) of nitrate
and ammonium, respectively.
In Period II (Nov–Dec), we find opposite results with quadratic parabola
distributions for nitrate-$NO_x$ response (Eq. 2) and ammonium-$NO_x$ response (Eq. 3),
but linearly increasing sulfate concentrations (average β values of +2.0% in BTH and
+2.6% in YRD; Fig. 9 and Fig. 10), leading to small $PM_{2.5}$ changes in winter. Such
nonlinear nitrate-$NO_x$ responses can be explained by the substantially increased
oxidants as we gradually reduce $NO_x$ emissions in each scenario. It is noted that in
winter the nitrate-$NO_x$ response highly depends on the production of $N_2O_5$, which is
produced from the $NO_2 \xrightarrow{O_3} NO_3 \xrightarrow{NO_2} N_2O_5$ chemical reactions and is a crucial intermediate
product for nitrate formation. Under the low $NO_x$ emission reduction conditions, the
production of $N_2O_5$ is more sensitive to the atmospheric oxidants concentrations. The
significant increases of surface $O_3$ in each $NO_x$ emission scenario in the VOC-poor
environment (Fig. 9(b, d)) lead to an enhancement of $N_2O_5$ levels from 10% to more
than 100%. In spite of the $HNO_3$ concentration remaining nearly unchanged or
decreasing slightly by less than 5% in response to $NO_x$ control, nitrate is found to
increase (average β values of +4.1% in BTH and +5.1% in YRD) with higher $N_2O_5$
produced from the increased ozone introduced by attenuated titration. An inflexion
point appears at the 40–50% $NO_x$ emission reduction scenario, and a further reduction
in $NO_x$ emissions is predicted to cause −10.5% and −5.3% reductions of surface
particulate nitrate and ammonium for BTH, and −7.7% and −7.4% for YRD.

These results reveal that the increase in atmospheric oxidants in response to $NO_x$

emission control can offset the decreasing precursors concentrations and further
enhance the formation of secondary nitrate, as recently found during the COVID-19
pandemic (Huang et al., 2020; Li et al., 2021b).
**3.3.3 Impacts of 2012–2016 $NO_x$ control strategy on particulate pollution**

During the 12$^{th}$ Five-Year Plan period (2011–2015), a series of end-of-pipe

pollutant controls (e.g., Selective Catalytic Reduction techniques) were carried out for
power, industry and transportation sectors. These measures effectively controlled the
national $NO_x$ emissions by 22.8% from 2012 to 2016 (MEIC v1.3) in China. To
quantify the effects of recent $NO_x$ control measures on the levels of photochemical
oxidants and particulate nitrate, we conduct an additional simulation with $NO_x$
emissions set to the levels of 2012 in E1.

The model simulations (Fig. 11) suggest that reducing China's $NO_x$ emissions

alone from 2012 to 2016 leads to an average −24.9%~−8.6% decrease of $NO_x$
concentrations in the surface layer. As previously pointed out, the 2012–2016 $NO_x$
emission control measures lead to increased $O_3$ and OH levels in winter, which offset
the effectiveness of $NO_x$ emission reduction in alleviating winter nitrate. No obvious
declines in the winter nitrate levels are observed and even increases in some areas
(+8.8% in BTH and 14.4% in YRD; Fig. S2–S3). As shown, the largest $PM_{2.5}$
responses shift towards the southern Hebei and central China provinces, where the
wintertime $PM_{2.5}$ concentrations are particularly high in this region. The substantial
emission changes from 2012 to 2016 lower the $PM_{2.5}$ air pollution by up to −1.8% in
BTH and −3.5% in YRD for Period I and oppositely increase the surface $PM_{2.5}$ by
2.4% in BTH and 4.7% in YRD for Period II. The past $NO_x$ emission control strategy
leads to increased atmospheric oxidants levels and deteriorated particulate pollution in
winter due to the nonlinear photochemistry and aerosol chemical feedbacks, without
regard to the other emission control measures. This conclusion is also supported by
evidence from the recent field observations (Fu et al., 2020).
**3.3.4 Responses of particulate nitrate to multi-pollutants cooperative controls**
In order to evaluate the effectiveness of multi-pollutants cooperative controls in
China, three series of additional simulations ($C_{S-N}$, $C_{N-N}$ and $C_{V-N}$) are also designed to
show the responses of nitrate and $PM_{2.5}$ pollution to the emission controls of $NO_x$,
$SO_2$, $NH_3$ and $VOC_s$, respectively. The results (Fig. 12) show that atmospheric $NH_3$
and $VOC_s$ are effective in controlling the particulate nitrate pollution for both seasons,
whereas decreasing the $SO_2$ and $NO_x$ emissions may have counter-intuitive effects on
the concentration levels of nitrate aerosols.
Atmospheric $NH_3$ acts as a critical neutralizing species for SIA production and
efficient haze mitigation (Liu et al., 2019). According to the WRF-Chem simulation,
reduction of $NH_3$ emissions may be effective in reducing the nitrate component, with
an average β value of −10.0% in BTH and −10.3% in YRD for Period I, and −8.3% in
BTH and −11.5% in YRD for Period II, primarily by suppressing the ammonium
nitrate formation. Quantitatively, a 10% reduction in $NH_3$ emissions can alleviate the
PM$_{2.5}$ pollution by −2.7% during summer-autumn and −3.2% during winter in the two
Chinese megacities. Atmospheric chemistry modeling by Wen et al. (2021) also
indicated that controlling NH$_3$ emissions in Beijing would significantly reduce the
population-weighted PM$_{2.5}$ concentrations by 6.2–21% with 60–100% NH$_3$ reductions
in January, implying the need to consider NH$_3$ emission controls when designing the
PM$_{2.5}$ pollution mitigation strategies.
VOC$_s$, which is not a direct precursor for SIA, is effective in SIA controls due to
their influences on the atmospheric oxidation cycles (Tsimpidi et al., 2008; Womack
et al., 2019; Nguyen and Dabdub, 2002). Our results suggest that decreasing VOC$_s$
emissions per 10% would suppress the oxidation formation of nitrate and decrease the
nitrate concentrations by −2.5% in BTH and −1.7% in YRD for Period I, and −5.0%
in BTH and −6.3% in YRD for Period II. The reduction of VOC$_s$ emissions would
result in a decrease of PM$_{2.5}$ by −0.7% during summer-autumn and −1.8% during
winter in the two megacities. Tsimpidi et al. (2008) also showed that the reduction of
VOC$_s$ emissions caused a marginal increase of PM$_{2.5}$ during summer in eastern United
States, whereas it resulted in a decrease of atmospheric oxidant levels and 5–20%
reduction of both inorganic and organic PM$_{2.5}$ components during winter. Larger and
synchronized NO$_x$ and VOC$_s$ emissions controls are required to overcome the adverse
effects of nonlinear photochemistry and aerosol chemical feedbacks.
The SO$_2$ emission reduction, although effective in reducing sulfate and PM$_{2.5}$, is
not successful in regulating the nitrate pollution due to the chemical competition in
nitrate and sulfate formations (Geng et al., 2017; Wang et al., 2013). Changes in
nitrate concentration are linearly associated with the SO$_2$ emission reductions, with
the average β values of 2.9% during summer-autumn and 1.3% during winter.
Decreasing SO$_2$ emissions is less effective (a β value of −0.7%) in mitigating the
wintertime haze pollution because that the benefit of $SO_2$ reduction is partly offset by
the significant increase of nitrate, demonstrating the critical role of multi-pollutants
cooperative controls. Lei et al. (2013) evaluated the impacts of $SO_2$ control strategies
on nitrate and sulfate production in USA and also found that the competition for bases
in nitrate and sulfate formation significantly affects the nitrate concentrations.
Our results emphasize that future nitrate and $PM_{2.5}$ pollution mitigation strategies
should focus on reducing the chemical precursors and key atmospheric oxidants
involved in the production of secondary aerosols. The recent "Three-year Action Plan
Fighting for a Blue Sky" calls for stringent emissions controls of $NO_x$, $SO_2$, $VOC_s$ and
$NH_3$ but without specific reduction targets. Such emission changes would emphasize
the need to jointly consider multi-pollutants emissions controls for mitigating haze air
pollution.
**4 Conclusions**
Recent air pollution actions have significantly lowered the $PM_{2.5}$ levels in China
via controlling emissions of $SO_2$ and $NO_x$, but raised a new question of how effective
the $NO_x$ emission controls can be on the mitigation of emerging nitrate and ozone air
pollution. We use comprehensive measurements and a regional meteorology-
chemistry model with optimized mechanisms to establish the nonlinear responses
between particulate nitrate and $NO_x$ emission controls in the megalopolises of China.
Nitrate is an essential component of $PM_{2.5}$ in eastern China, accounting for 9.4–
15.5% and 11.5–32.1% of the total $PM_{2.5}$ mass for the warm and cold seasons,
respectively. We find that the efficiency of $PM_{2.5}$ reduction is highly sensitive to $NO_x$
emissions and it varies in different seasons depending on the ozone chemical regimes.
The reduction of $NO_x$ emissions results in almost linearly lower $PM_{2.5}$ by −2.2% in
BTH and −2.9% in YRD per 10% cut of $NO_x$ emissions during summer-autumn,
whereas it increases the atmospheric oxidants levels and leads to a rather complicated
response of the PM components in winter. Nitrate is found to increase (average β
values of +4.1% in BTH and +5.1% in YRD) in winter with higher $N_2O_5$ intermediate
produced from the increased ozone introduced by attenuated titration, despite the
nearly unchanged or slightly decreased $HNO_3$ concentrations in response to $NO_x$
control. An inflexion point appears at 40–50% $NO_x$ emission reduction, and a further
reduction of $NO_x$ emissions is predicted to cause −10.5% reductions of particulate
nitrate for BTH and −7.7% for YRD. In addition, the 2012–2016 $NO_x$ emission
control strategy leads to −24.9%~−8.6% decreases of surface $NO_x$ concentrations, and
no changes or even increases of wintertime nitrate in BTH (+8.8%) and YRD (14.4%).
Our results also emphasize that atmospheric $NH_3$ and $VOC_s$ are effective in
controlling the particulate nitrate pollution, whereas decreasing the $SO_2$ and $NO_x$
emissions may have counter-intuitive effects on nitrate aerosols. These results provide
insights for developing mitigation strategies for the ubiquitous nitrate aerosols in
winter haze of China.
**Author contribution**
Mengmeng Li developed the model code, designed the numerical experiments,
and wrote the original draft. Zihan Zhang carried out the numerical experiments.
Quan Yao provided and analyzed some of the data. Min Xie, Shu Li and Bingliang
Zhuang validated and analyzed the model results. Tijian Wang and Yong Han
reviewed and revised the manuscript.
**Competing interests**
The authors declare that they have no conflict of interest.
**Acknowledgement**
This study is funded by the National Natural Science Foundation of China
(41975153, 42077192 and 41775026), the National Key Basic Research Development
Program of China (2019YFC0214603, 2020YFA0607802), and the Emory
University-Nanjing University Collaborative Research Grant.
**Data availability statement**
The      WRF-Chem     model     version     4.1     is     available     at
http://www2.mmm.ucar.edu/wrf/users/downloads.html. The NCEP FNL data are
accessible at the National Center for Atmospheric Research (NCAR) Research Data
Archive (RDA; http://rda.ucar.edu/datasets/ds083.2/). The MEIC anthropogenic
emission inventories are available at www.meicmodel.org, and for more information,
please contact Q. Zhang (qiangzhang@tsinghua.edu.cn). The surface weather data are
accessible at the Integrated Surface Database (https://www.ncdc.noaa.gov/isd/data-
access). The surface air pollutants and aerosol species data are provided by Chinese
National Environmental Monitoring Center (http://www.cnemc.cn/en/) and archived
at https://doi.org/10.6084/m9.figshare.12818807.v1.

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

Contributions of Major Oxidation Pathways to PM2.5 Nitrate Formation in
Beijing, J Geophys Res-Atmos, 124, 4174-4185, 2019.
Wen, L., Xue, L. K., Wang, X. F., Xu, C. H., Chen, T. S., Yang, L. X., Wang, T.,
Zhang, Q. Z., and Wang, W. X.: Summertime fine particulate nitrate pollution in
the North China Plain: increasing trends, formation mechanisms and implications
for control policy, Atmos Chem Phys, 18, 11261-11275, 2018.
Wen, Z., Xu, W., Pan, X. Y., Han, M. J., Wang, C., Benedict, K., Tang, A. H., Collet,
J. L., and Liu, X. J.: Effects of reactive nitrogen gases on the aerosol formation
in Beijing from late autumn to early spring, Environ Res Lett, 16, 025005, doi:
10.1088/1748-9326/abd973, 2021.
Womack, C. C., McDuffie, E. E., Edwards, P. M., Bares, R., de Gouw, J. A.,
Docherty, K. S., Dube, W. P., Fibiger, D. L., Franchin, A., Gilman, J. B.,
Goldberger, L., Lee, B. H., Lin, J. C., Lone, R., Middlebrook, A. M., Millet, D.
B., Moravek, A., Murphy, J. G., Quinn, P. K., Riedel, T. P., Roberts, J. M.,
Thornton, J. A., Valin, L. C., Veres, P. R., Whitehill, A. R., Wild, R. J., Warneke,
C., Yuan, B., Baasandorj, M., and Brown, S. S.: An Odd Oxygen Framework for
Wintertime Ammonium Nitrate Aerosol Pollution in Urban Areas: NOx and
VOC Control as Mitigation Strategies, Geophys Res Lett, 46, 4971-4979, 2019.
Xie, M., Zhu, K. G., Wang, T. J., Yang, H. M., Zhuang, B. L., Li, S., Li, M. G., Zhu,
X. S., and Ouyang, Y.: Application of photochemical indicators to evaluate
ozone nonlinear chemistry and pollution control countermeasure in China,
Atmos Environ, 99, 466-473, 2014.
Xue, J., Yuan, Z. B., Lau, A. K. H., and Yu, J. Z.: Insights into factors affecting
nitrate in PM2.5 in a polluted high NOx environment through hourly

observations and size distribution measurements, J Geophys Res-Atmos, 119, 4888-4902, 2014.

Xue, J., Yuan, Z. B., Griffith, S. M., Yu, X., Lau, A. K. H., and Yu, J. Z.: Sulfate Formation Enhanced by a Cocktail of High NOx, SO2, Particulate Matter, and Droplet pH during Haze-Fog Events in Megacities in China: An Observation-Based Modeling Investigation, Environ Sci Technol, 50, 7325-7334, 2016.

Zaveri, R. A. and Peters, L. K.: A new lumped structure photochemical mechanism for large-scale applications, J Geophys Res-Atmos, 104, 30387-30415, 1999.

Zaveri, R. A., Easter, R. C., Fast, J. D., and Peters, L. K.: Model for Simulating Aerosol Interactions and Chemistry (MOSAIC), J Geophys Res-Atmos, 113, D13204, doi: 10.1029/2007jd008782, 2008.

Zhai, S. X., Jacob, D. J., Wang, X., Shen, L., Li, K., Zhang, Y. Z., Gui, K., Zhao, T. L., and Liao, H.: Fine particulate matter (PM2.5) trends in China, 2013-2018: separating contributions from anthropogenic emissions and meteorology, Atmos Chem Phys, 19, 11031-11041, 2019.

Zhang, W. Q., Tong, S. R., Ge, M. F., An, J. L., Shi, Z. B., Hou, S. Q., Xia, K. H., Qu, Y., Zhang, H. X., Chu, B. W., Sun, Y. L., and He, H.: Variations and sources of nitrous acid (HONO) during a severe pollution episode in Beijing in winter 2016, Sci Total Environ, 648, 253-262, 2019.

Zhang, Y. M., Wang, Y. Q., Zhang, X. Y., Shen, X. J., Sun, J. Y., Wu, L. Y., Zhang, Z. X., and Che, H. C.: Chemical Components, Variation, and Source Identification of PM1 during the Heavy Air Pollution Episodes in Beijing in December 2016, J Meteorol Res-Prc, 32, 1-13, 2018.

Zhao, M. F., Xiu, G. L., Qiao, T., Li, Y. L., and Yu, J. Z.: Characteristics of Haze Pollution Episodes and Analysis of a Typical Winter Haze Process in Shanghai, Aerosol Air Qual Res, 16, 1625-1637, 2016.

Zhao, P. S., Dong, F., He, D., Zhao, X. J., Zhang, X. L., Zhang, W. Z., Yao, Q., and Liu, H. Y.: Characteristics of concentrations and chemical compositions for PM2.5 in the region of Beijing, Tianjin, and Hebei, China, Atmos Chem Phys, 13, 4631-4644, 2013.

Zheng, B., Tong, D., Li, M., Liu, F., Hong, C. P., Geng, G. N., Li, H. Y., Li, X., Peng,

784          L. Q., Qi, J., Yan, L., Zhang, Y. X., Zhao, H. Y., Zheng, Y. X., He, K. B., and

Zhang, Q.: Trends in China's anthropogenic emissions since 2010 as the

consequence of clean air actions, Atmos Chem Phys, 18, 14095-14111, 2018.


**Table 1**. The emission scenarios in WRF-Chem numerical experiments

| Simulation scenarios | Descriptions |
|---|---|
| B0 | Base simulation under the 2016 emission conditions. |
| $C_N$ (N=1/2/…/8) | Same as B0, but anthropogenic $NO_x$ emissions are reduced by 10%, 20%...80%, respectively, relative to the usual levels in 2016. |
| $C_{S-N}$ (N=2/4/6/8) | Same as B0, but anthropogenic $SO_2$ emissions are reduced by 20%, 40%...80%, respectively, relative to the usual levels in 2016. |
| $C_{N-N}$ (N=2/4/6/8) | Same as B0, but anthropogenic $NH_3$ emissions are reduced by 20%, 40%...80%, respectively, relative to the usual levels in 2016. |
| $C_{V-N}$ (N=2/4/6/8) | Same as B0, but anthropogenic $VOC_s$ emissions are reduced by 20%, 40%...80%, respectively, relative to the usual levels in 2016. |
| N0 | Same as B0, but only consider the $NO_2+OH$ gas-phase oxidation pathway for the production of nitrate aerosol. |
| E1 | Same as B0, but anthropogenic $NO_x$ emissions are replaced using the MEIC inventory in 2012. |


**Table 2**. Statistical evaluations of the model meteorological performance

| Variable | Obs | Sim | $R$ [a] | MB [a] | NMB [a] | ME [a] | RMSE [a] |
|---|---|---|---|---|---|---|---|
| Period I (15 August to 16 September) | | | | | | | |
| Temperature (°C) | 24.04 | 23.91 | 0.89 | −0.13 | −0.55% | 1.98 | 2.63 |
| Humidity (%) | 70.89 | 66.88 | 0.78 | −4.01 | −5.65% | 11.07 | 14.67 |
| Wind speed (m s$^{-1}$) | 2.46 | 3.04 | 0.50 | 0.58 | 23.72% | 1.38 | 1.83 |
| Period II (24 November to 26 December) | | | | | | | |
| Temperature (°C) | 3.43 | 3.40 | 0.94 | −0.03 | −0.80% | 2.18 | 2.83 |
| Humidity (%) | 69.85 | 65.27 | 0.63 | −4.58 | −6.56% | 13.51 | 17.88 |
| Wind speed (m s$^{-1}$) | 2.61 | 3.66 | 0.55 | 1.06 | 40.64% | 1.70 | 2.23 |

[a] $R$: correlation efficient; MB: mean bias; NMB: normalized mean bias; ME: mean
error; RMSE: root mean square error.


**Table 3**. Statistical evaluations of the model chemical performance

| Variable | Obs | Sim | MB | NMB | Obs | Sim | MB | NMB |
|---|---|---|---|---|---|---|---|---|
| | | Period I | | | | Period II | | |
| $PM_{2.5}$ | 36.88 | 33.22 | −3.66 | −9.92% | 91.59 | 64.28 | −27.31 | −29.82% |
| $SO_2$ | 17.65 | 16.51 | −1.14 | −6.46% | 41.45 | 29.80 | −11.65 | −28.11% |
| $NO_2$ | 28.53 | 33.23 | 4.70 | 16.47% | 53.01 | 54.28 | 1.27 | 2.40% |
| Daily-maximum $O_3$ | 237.45 | 255.77 | 18.32 | 7.72% | 125.62 | 86.61 | −39.01 | −31.05% |

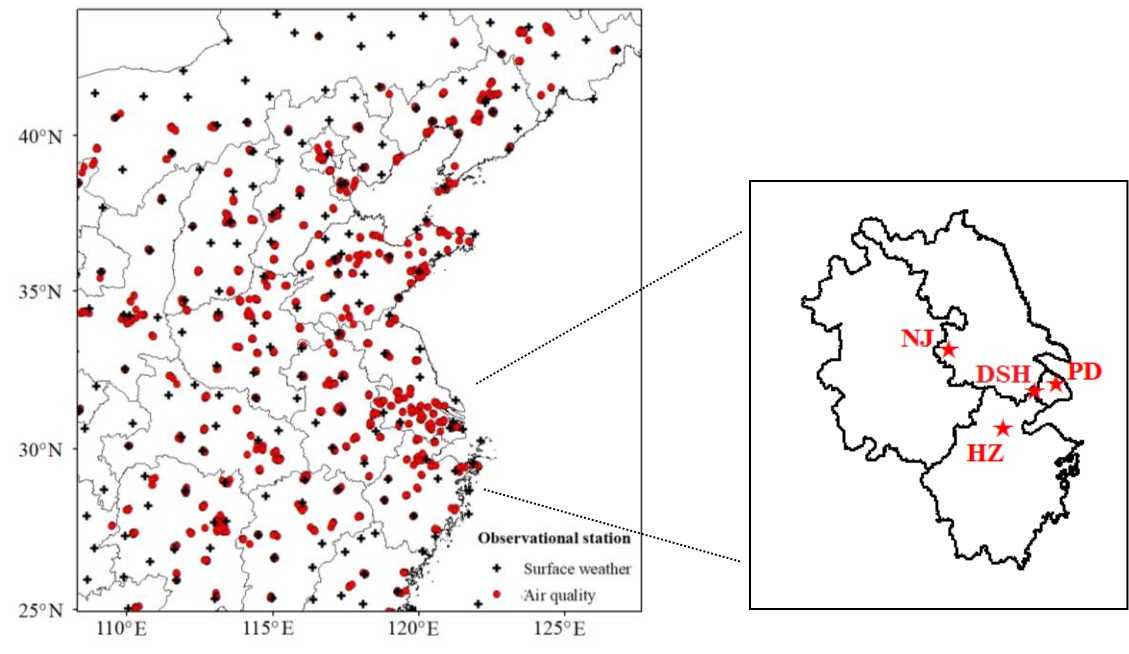

**Fig. 1**. WRF-Chem domain configuration and observational stations. Black crosses:
surface weather stations; Red dots: CNEMC routine air quality monitoring stations;
Red stars: surface supersites in YRD.

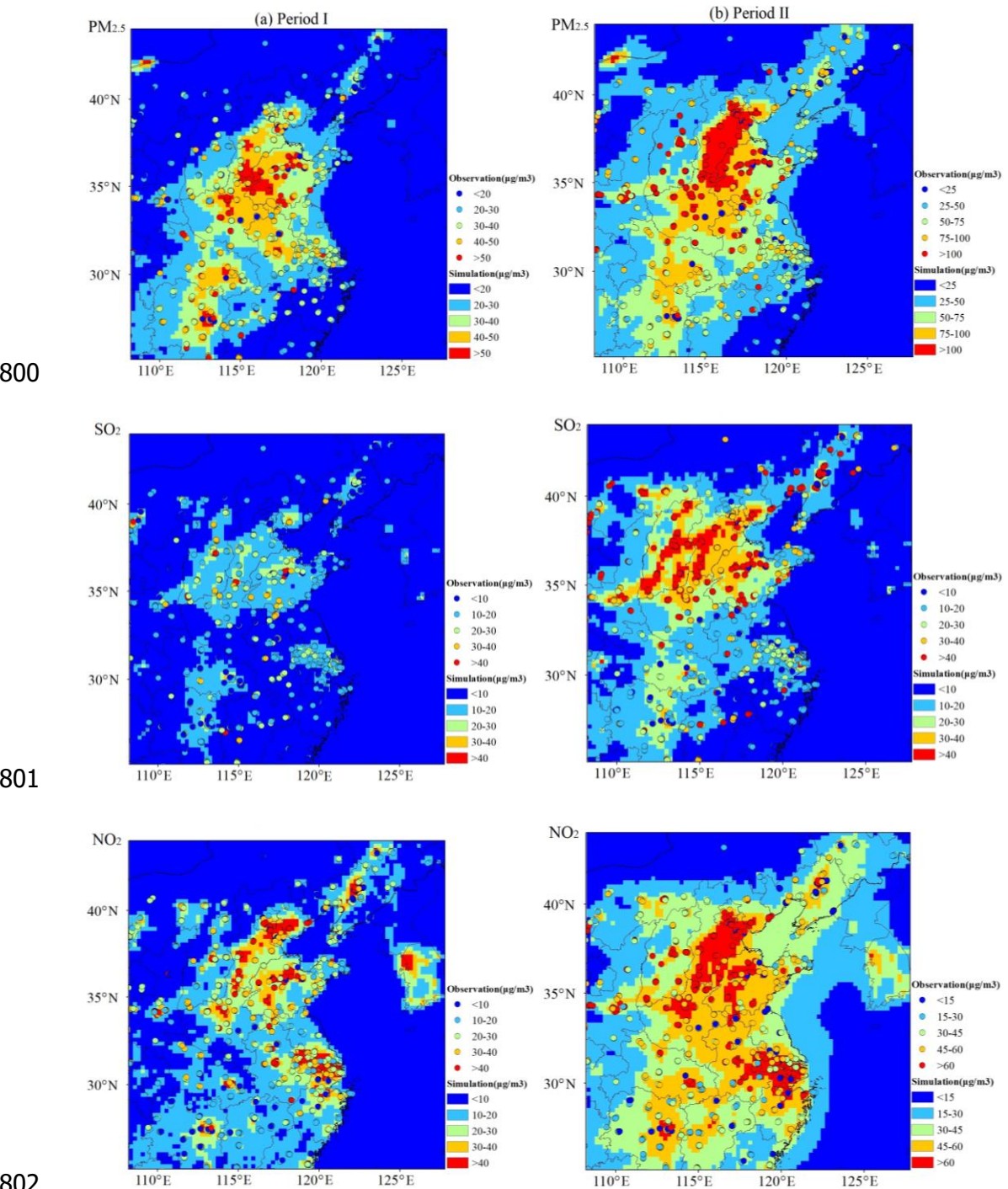




Fig. 2. Spatial patterns of the surface average PM$_{2.5}$, NO$_2$, SO$_2$ and daily-maximum O$_3$ concentrations in Period I (left panels) and Period II (right panels) from the WRF-Chem modeling (shaded contours) and routine air quality observations (dots).

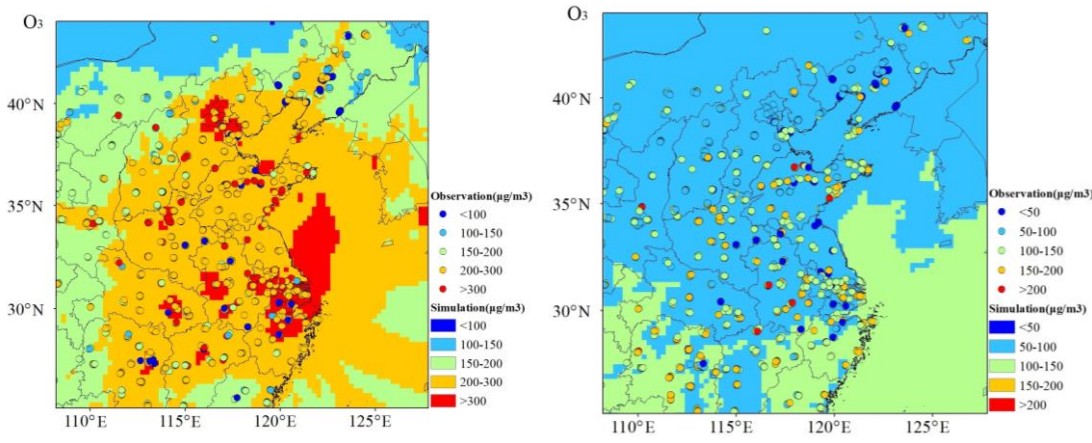


**Fig. 2**. Continued.

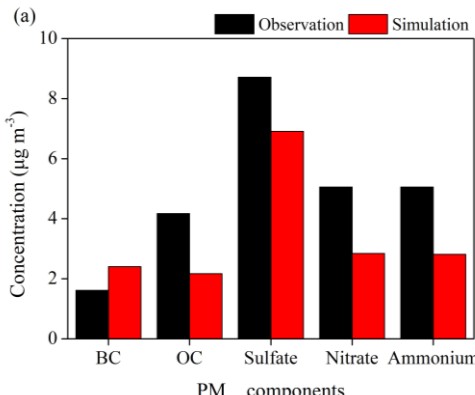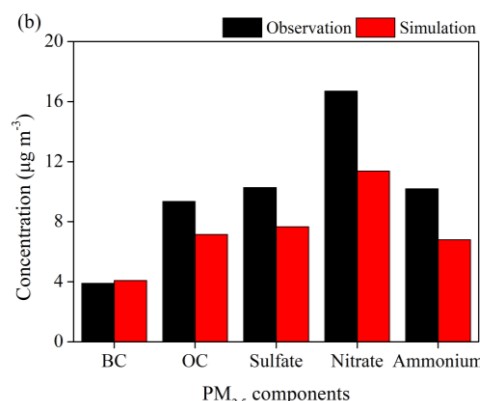


**Fig. 3**. Comparisons of surface PM₂.₅ components from WRF-Chem simulations and
observations in Period I (a) and Period II (b) at the four supersites in YRD.


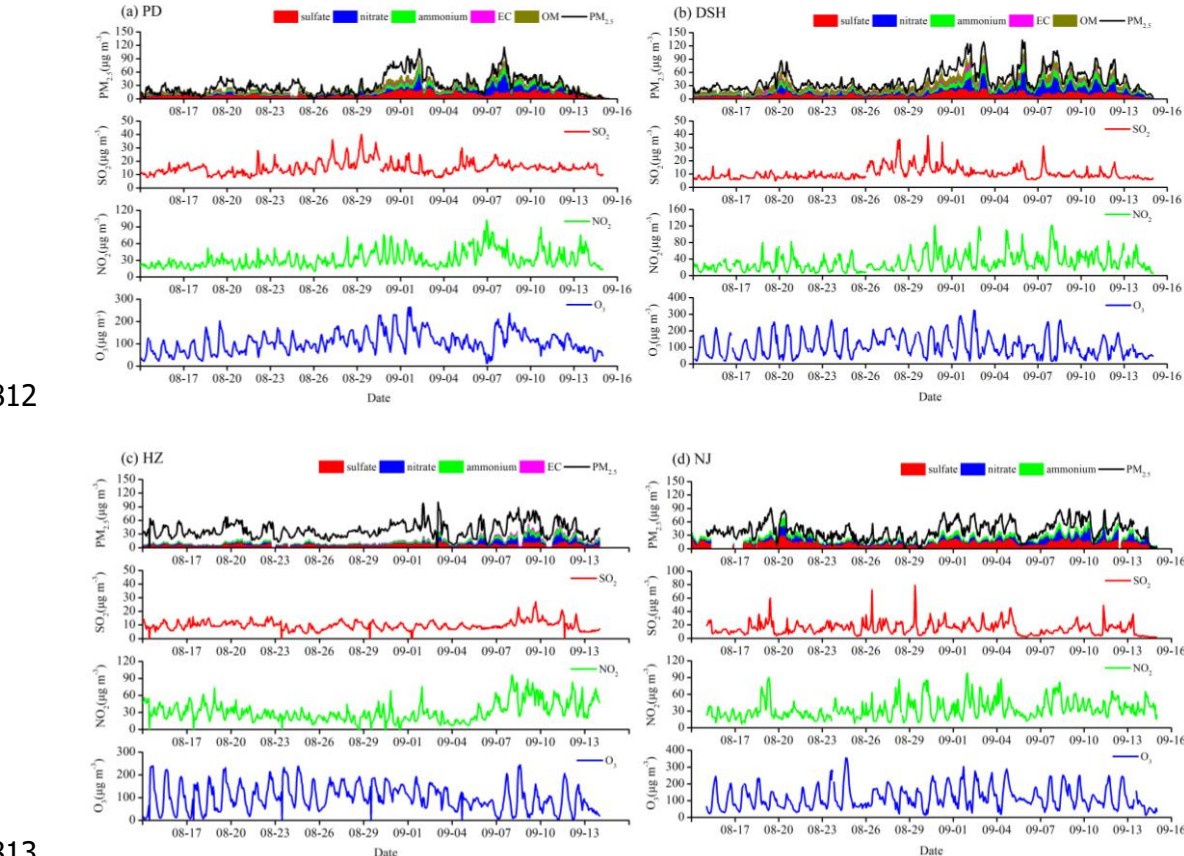


**Fig. 4**. Observed aerosol composition and gaseous pollutants concentrations at the
four supersites during Period I.

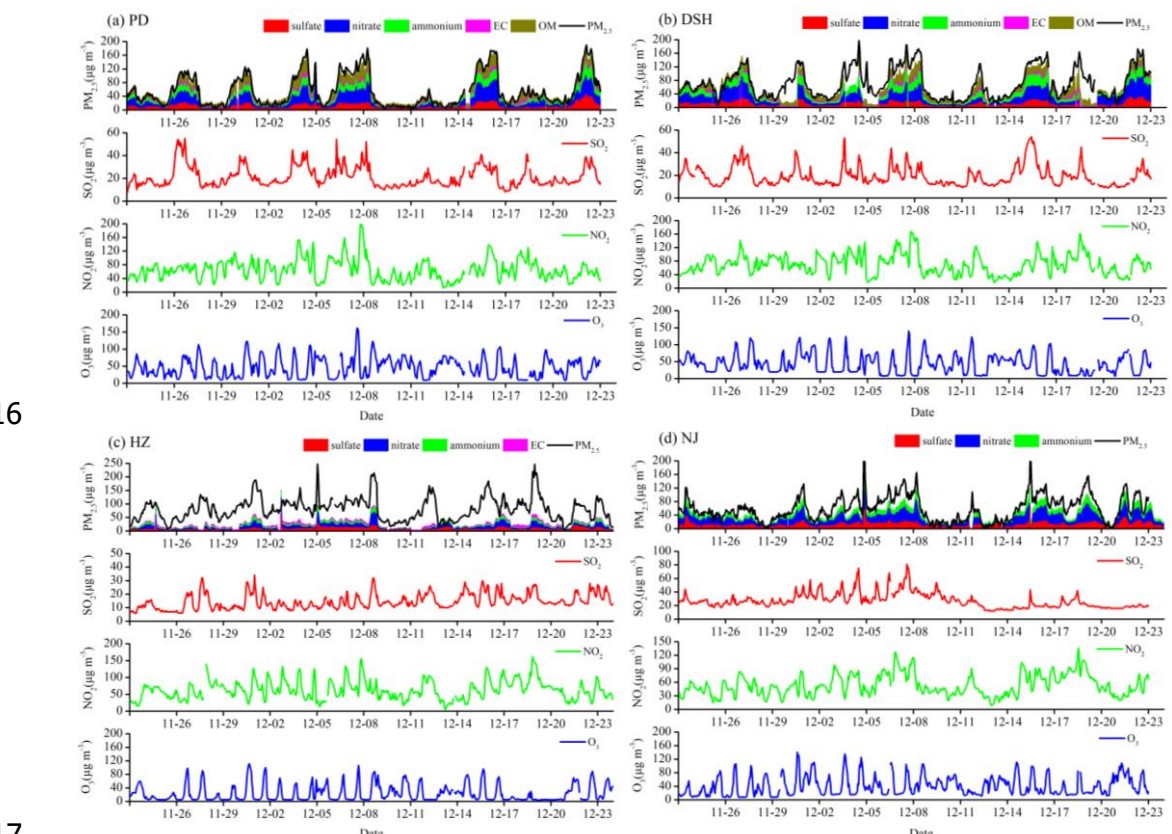


**Fig. 5**. Same as Fig. 4, but for Period II.

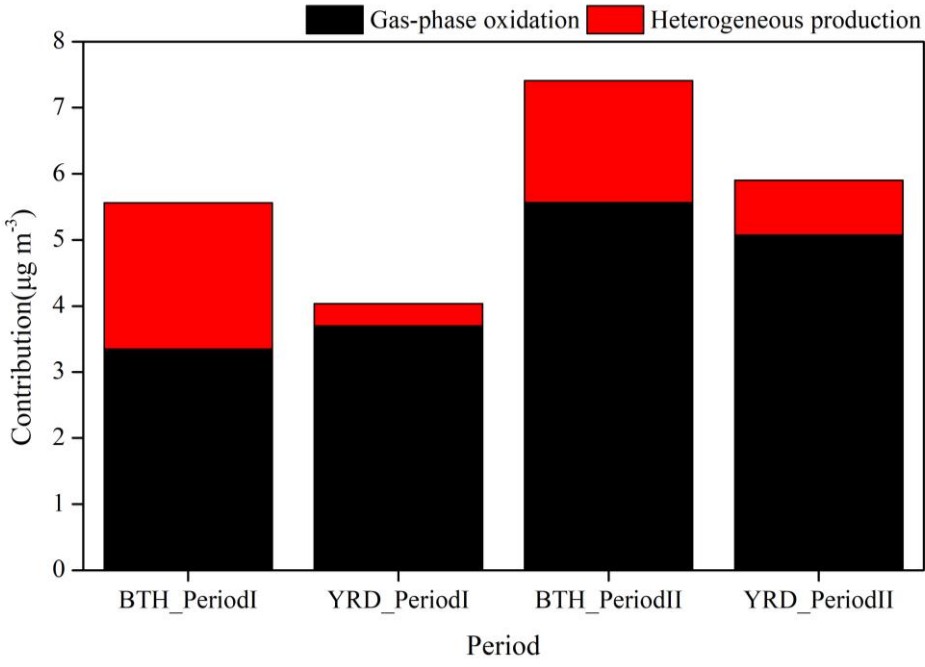


**Fig. 6**. Contributions of gas-phase oxidation and heterogeneous production to the
surface nitrate concentrations for the BTH and YRD regions in two seasons.

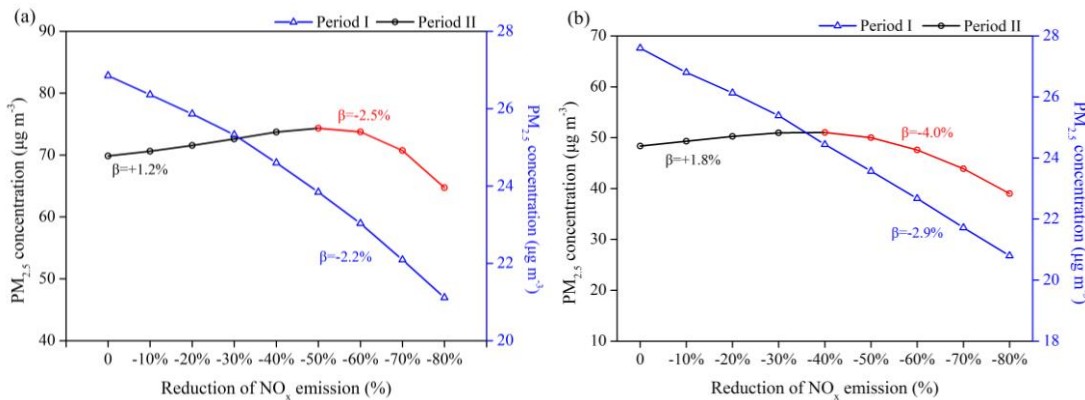


**Fig. 7**. Responses of surface $PM_{2.5}$ concentrations to the $NO_x$ emission reduction
scenarios in (a) BTH and (b) YRD. The calculated $NO_x$ emission control efficiency (β)
is also marked in the figure.


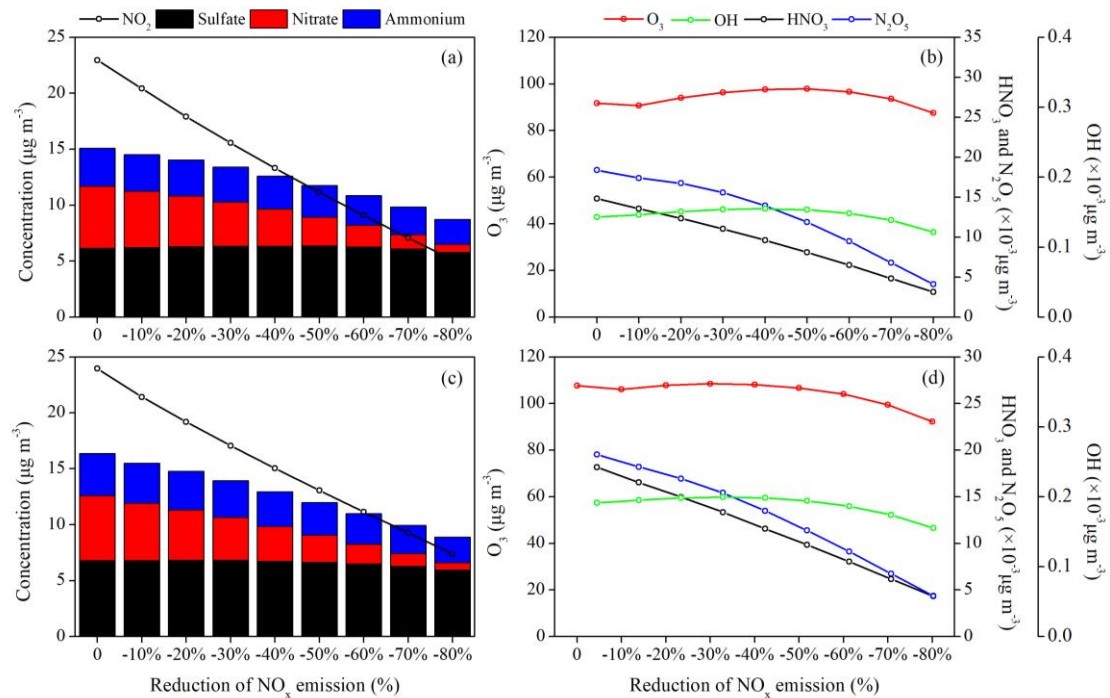


Fig. 8. Responses of the surface concentrations of SIA components and key atmospheric trace gases ($NO_2$, $O_3$, OH, $HNO_3$ and $NO_3$) to the $NO_x$ emission reduction scenarios in (a, b) BTH and (c, d) YRD during Period I.

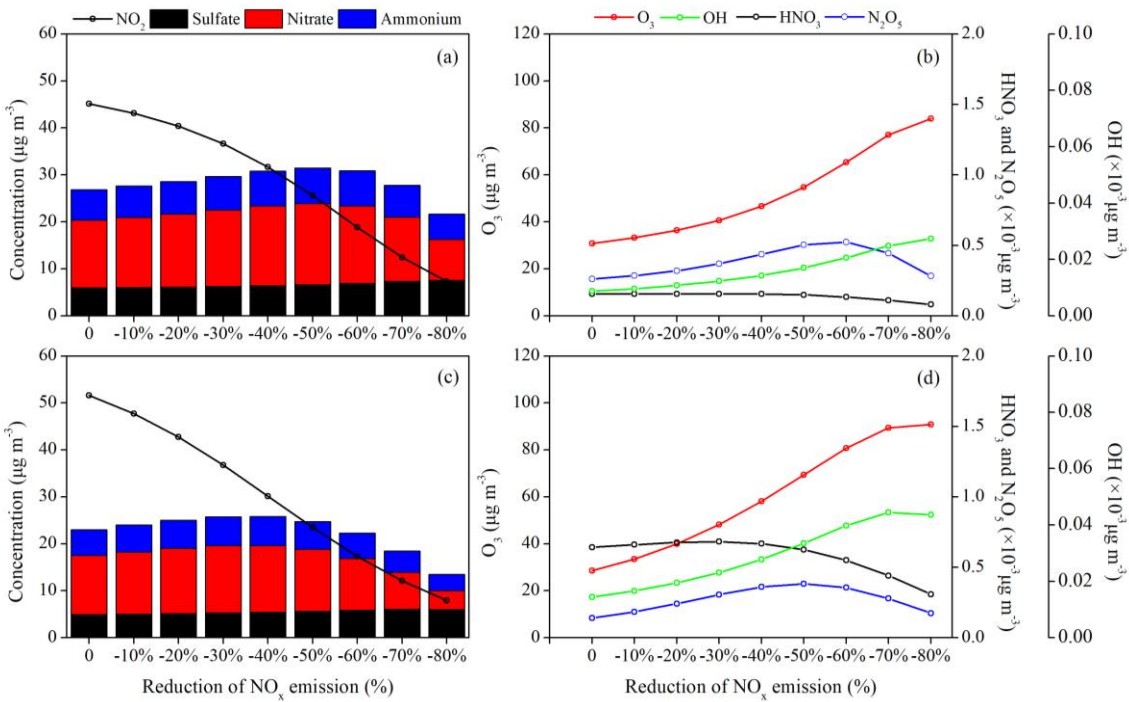


Fig. 9. Same as Fig. 8, but for Period II.

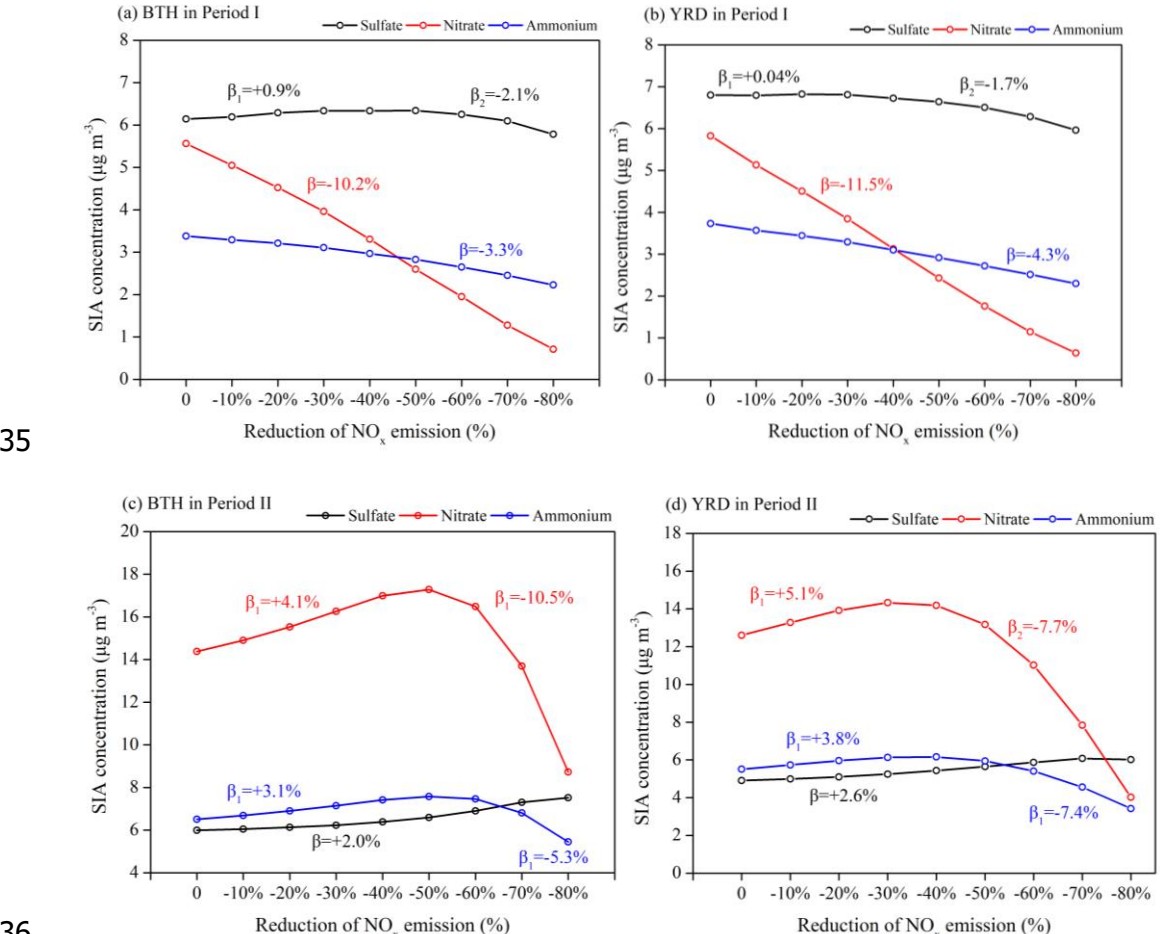



**Fig. 10**. Responses of the surface concentrations of SIA components to the NO$_x$ emission reduction scenarios and their emission control efficiencies in (a, b) Period I and (c, d) Period II.

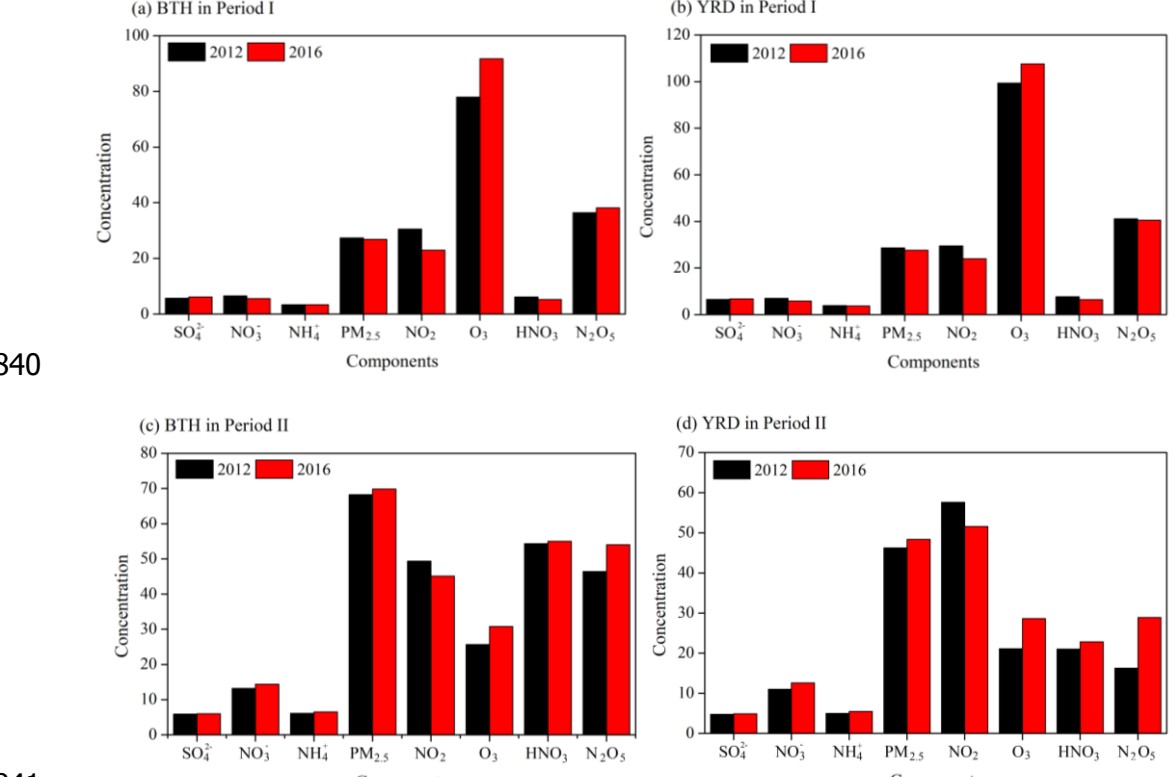



**Fig. 11**. Changes in the concentrations of surface PM$_{2.5}$, SIA components and key
atmospheric trace (NO$_2$, O$_3$, HNO$_3$ and N$_2$O$_5$) due to the 2012–2016 NO$_x$ emission
reductions in China estimated as the differences between the base simulation and E1
scenario. The units are ppt for HNO$_3$ and N$_2$O$_5$, and μg m$^{-3}$ for other chemical species.

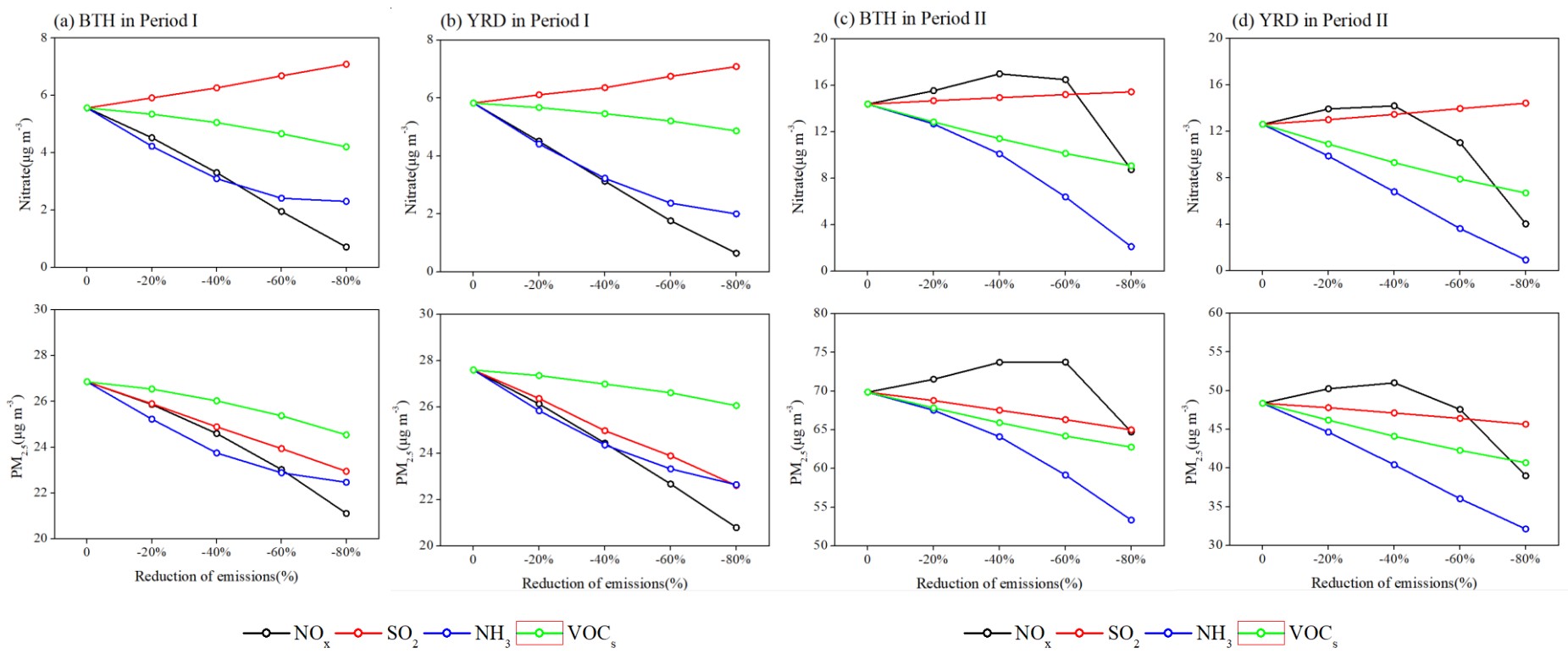



**Fig. 12**. Responses of the surface nitrate (upper panels) and PM$_{2.5}$ (bottom panels) concentrations to the emission reduction scenarios of NO$_x$,
SO$_2$, NH$_3$ and VOC$_s$ during Period I (a, b) and Period II (c, d).