# Peer review of "Nonlinear responses of particulate nitrate to $NO_x$ emission controls in the megalopolises of China"

_Atmospheric Chemistry and Physics, 2021_

## Author Comment (AC1)

**Response to Reviewer1**

**General comments:**

*Li et al. present a model analysis of secondary inorganic aerosol (SIA) in major cities of Eastern China. They show data of PM$_{2.5}$ from a network across China from 2012 to 2016 and speciated PM$_{2.5}$ from major supersites in Eastern China. They analyze the response of sulfate and nitrate to the reductions in NO$_x$ that have occurred in the 4 year period, and to a theoretical series of reductions spanning 10 to 80%. Due to the response of oxidants to NO$_x$, the changes in SIA are approximately linear in summer but highly non-linear in winter, indicating NO$_x$ saturation of oxidant production in winter.*

*The results are relevant to the understanding of air pollution and its recent trends in China and is of interest to the readership of ACP. **Publication is recommended following attention to the specific comments below**.*

**Response:** The paper has been revised according to the reviewer's comments. For details, see the line-by-line responses below.

**Specific Comments:**

*Line 71: Neither of the stated assumptions, stagnant atmosphere or a fixed NO$_2$/NO$_y$ ratio, is a condition for a box model. Suggest eliminating this statement.*

**Response: Accepted.** This sentence is rephrased as "…the basic hypotheses in box models to predict nitrate production are also unreasonable in the real atmosphere…". For details, see *Line 73-74*.

*Line 77-78: Is the NO$_x$ sensitivity of Chinese ozone a well accepted result? Increasing O$_3$ trends have been attributed, for example, to effects associated with decreasing PM$_{2.5}$, e.g., Li, K., D.J. Jacob, H. Liao, L. Shen, Q. Zhang, and K.H. Bates, Anthropogenic drivers of 2013–2017 trends in summer surface ozone in China. Proceedings of the National Academy of Sciences, 2019. **116**(2): p. 422.*

**Response: Accepted.** More references are supplemented to support the claim that the production of O$_3$ is usually limited by VOC$_s$ in Chinese cities (Xie et al., 2014; Dong et al., 2014; Liu et al., 2010). Li et al. (2021) and Liu and Wang (2020) examined the influencing factors on surface O$_3$ trends in China from 2013 to 2017 using regional models. They highlighted that the control of NO$_x$ emissions explained 11–35% of increased O$_3$ due to the nonlinear NO$_x$-VOC$_s$-O$_3$ chemistry, and that for most regions the magnitudes could be comparable to those resulting

from the meteorological influences and aerosol effects. For details, see *Line 77-86.*

*Line 248: It appears that the oxidation ratio of sulfur decreases, but nitrogen increases, between the warm and cold seasons. Is that correct? In this case, there is more going on than just enhanced secondary oxidant formation. There is also a change in chemical mechanism between S and N.*

**Response: Accepted.** The seasonal distinctions for SOR and NOR, i.e., higher SOR in summer-autumn than winter and opposite for NOR, have been reported by Shu et al. (2019). They attributed the weakened conversion from $NO_2$ to nitrate in summer to the volatility and evaporative loss of nitrate (Sun et al., 2012). The sharp increase of particles and moderate ambient humidity in winter also benefit the heterogeneous formation of SIA, leading to high NOR and SOR (Wang et al., 2012). For details, see *Line 267-274.*

*Line 249: Here and above, specify more clearly what is meant by the two pathways – presumably this is the difference between OH + $NO_2$ (gas phase) and $N_2O_5$ hydrolysis (heterogeneous), but it is not clear from this text or figure 6 what this specifically means.*

**Response: Accepted.** The two pathways represent the $NO_2$+OH gas-phase oxidation and heterogeneous reactions for nitrate production calculated from B0 and E0 simulations. It is specified clearly in the figure captions and man text. For details, see *Line 148-153* and *Line 275-283.*

*Line 258-259: Two comments. First, the global burden of nitrate production pathways may, or may not, be relevant at a regional level. Second, there are more updated papers indicating that the two pathways considered here are more equal at the global scale.*

*Alexander, B., T. Sherwen, C.D. Holmes, J.A. Fisher, Q. Chen, M.J. Evans, and P. Kasibhatla, Global inorganic nitrate production mechanisms: comparison of a global model with nitrate isotope observations. Atmos. Chem. Phys., 2020. **20**(6): p. 3859-3877.*

**Response: Accepted.** We extend the discussions about the formation pathways for nitrate calculated in the WRF-Chem model and their comparisons with recent assessments in China and worldwide. The calculated contributions (60.19–91.71% for $NO_2$+OH oxidation and 8.29–39.81% for heterogeneous pathways) are in line with previous assessments in China and globally (Alexander et al., 2009; Alexander et al., 2020; Wang et al., 2019; Wang et al., 2017; Luo et al., 2021; Liu et al., 2020; He et al., 2020). The nitrate formation from heterogeneous

reactions is moderately underestimated in this study, possibly due to the uncertainties of heterogeneous uptake coefficients and unclear reaction mechanisms applied in the model (Li et al., 2019; Xue et al., 2016; He et al., 2014). For details, see *Line 283-297*.

*Line 275: The responses do not appear "bell shaped".   Is there a better descriptor?*

**Response: Accepted.** The response is described as "quadratic parabola distribution". For details, see *Line 314*.

*Line 301-319: The changes in sulfate in Figures 8 and 9 is quite difficult to discern, so it is not easy to match this text to the changes in the figure. Can sulfate be plotted on its own scale to illustrate these changes?*

*Line 329-330: Same comment for sulfate. Changes are very difficult to see.*

**Response: Accepted.** The responses of SIA components to the $NO_x$ emission controls are plotted in *Fig. 10* of the revised manuscript.

[Figure]

**Fig. 10**. Responses of the surface concentrations of SIA components to the $NO_x$ emission reduction scenarios and their emission control efficiencies in (a, b) Period I

and (c, d) Period II.

*Also, the small dependence of sulfate on NO$_x$ indicates that OH is also not a strong function of NO$_x$ in this model, but later the OH changes are quoted as being large compared to the changes in sulfate. If OH is the most important factor for sulfate changes, why are the modeled sulfate changes so much smaller than the changes in OH?*

**Response: Accepted.** From the response curves of OH and sulfate (*Fig. 8-10*), it is seen that the sulfate-NO$_x$ response pattern is similar to that of OH-NO$_x$ but with smaller changes. It is because that the concentration levels of sulfate are not only limited by the oxidants levels in the atmosphere but also many other factors, e.g., the abundance of its precursors (e.g., SO$_2$, NH$_3$), the chemical competition in sulfate and nitrate formation, and the heterogeneous uptake rates etc. (Dong et al., 2014; Meng et al., 1997). For details, see *Line 343-350* and *Line 376-377*.

**References:**

[revised manuscript text omitted]

---

## Author Comment (AC2)

**Response to Reviewer2**

**General comments:**

*This paper investigates the secondary inorganic aerosol formation in cities in China focusing on nitrate aerosol, using observed data from a Chinese network of station and a WRF-Chem mesoscale model to analyze the response of PM$_{2.5}$ to NO$_x$ emission changes on a seasonal basis. The main findings are the almost linear response of secondary inorganic nitrogen to NO$_x$ emissions changes during summer while in winter the system is highly non-linear. Sensitivities to ammonia and sulfur dioxide emissions, which have been shown to be important, are discussed only based on existing literature. **The paper is within the scope of ACP and could be suitable for publication in the journal after a number of improvements suggested below**.*

**Major comments:**

*My major comment is that the authors miss to show a clear view of the interplay between O$_3$, SO$_2$, NO$_x$, NH$_3$ and PM$_{2.5}$. The authors provide some information in this respect in the conclusion but the manuscript can be largely improved in this respect. Therefore, I would suggest that the authors check on relevant references and integrate some of them in the introduction or discussion session. Some are listed below but more might be found by a literature search.*

Response: **Accepted.** Several improvements are performed in the revised manuscript as below:

(1) To show the responses of nitrate and PM$_{2.5}$ pollution to the cooperative controls of NO$_x$, SO$_2$, NH$_3$ and VOC$_s$, three series of additional simulations (see *Line 160-165* and *Table 1*; C$_{S-N}$, C$_{N-N}$ and C$_{V-N}$) are conducted with the anthropogenic emissions of SO$_2$, NH$_3$ and VOC$_s$ in China cut by 20%, 40%...and 80%. Our results emphasize that atmospheric NH$_3$ and VOC$_s$ are effective in controlling the particulate nitrate pollution, whereas decreasing the SO$_2$ and NO$_x$ emissions may have counter-intuitive effects on nitrate aerosols. For details, see *Sec. 3.3.4*.

(2) More relevant references are supplemented in the introduction section to explain the interplay between O$_3$, NO$_x$ and PM$_{2.5}$. For details, see *Line 77-86*.

(3) We extend the discussions about the formation pathways for nitrate calculated in the WRF-Chem model and their comparisons with recent assessments in China and worldwide. The calculated contributions (60.19–91.71% for NO$_2$+OH oxidation and 8.29–39.81% for heterogeneous pathways) are in line

with previous assessments in China and globally (Alexander et al., 2009; Alexander et al., 2020; Wang et al., 2019; Wang et al., 2017; Luo et al., 2021; Liu et al., 2020; He et al., 2020). For details, see *Line 148-153* and *Line 283-297*.

**Specific Comments:**

*Line 189: capture*

**Response:** **Accepted.** For details, see *Line 205*.

*Line 194: MB*

**Response:** **Accepted.** This sentence has been rephrased as "The model statistical evaluations show a mean bias (MB)…". For details, see *Line 210-211*.

*Line 211-212: that the consideration of the optimized… reduces…*

**Response:** **Accepted.** For details, see *Line 228*.

*Lines 228: was more than …*

**Response:** **Accepted.** For details, see *Line 246*.

*Line 232: (because OM has more than C, O and H) I suggest adding 'mainly' for the hydrogen and oxygen*

**Response:** **Accepted.** For details, see *Line 250*.

*Line 235: could you provide standard deviation for percent fractions ?*

**Response:** **Accepted.** We give the ranges for the percent fractions. This sentence is rephrased as "The three SIA components together account for 32.3–57.4% (48.6% on average) and 27.7–70.9% (56.9% on average) of the total $PM_{2.5}$ mass…". For details, see *Line 253-254*.

*Lines 236: SIA are the dominant components of $PM_{2.5}$ – 48.6% is not the majority – it is below 50%*

**Response:** **Accepted.** This sentence is rephrased as "…become the primary components of $PM_{2.5}$ in the two periods". For details, see *Line 255*.

*Lines 237-242: please explain the drivers, are temperature, liquid water content of the aerosol impacting these ratios*

**Response: Accepted.** The seasonal distinctions for sulfur and nitrogen oxidation ratios have also been reported by Shu et al. (2019). They attributed the weakened conversion from $NO_2$ to nitrate in summer to the volatility and evaporative loss of nitrate (Sun et al., 2012). The sharp increase of particles and moderate ambient humidity in winter also benefit the heterogeneous formation of SIA, leading to high NOR and SOR (Wang et al., 2012). For details, see *Line 267-274*.

*Lines 258-259: discussion could profit from additional literature – also heterogeneous reactions on dust can produce nitrate aerosol (see references above)*

**Response: Accepted.** We extend the discussions about the formation pathways for nitrate calculated in the WRF-Chem model and their comparisons with recent assessments in China and worldwide. The calculated contributions (60.19–91.71% for $NO_2$+OH oxidation and 8.29–39.81% for heterogeneous pathways) are in line with previous assessments in China and globally (Alexander et al., 2009; Alexander et al., 2020; Wang et al., 2019; Wang et al., 2017; Luo et al., 2021; Liu et al., 2020; He et al., 2020). The nitrate formation from heterogeneous reactions is moderately underestimated in this study, possibly due to the uncertainties of heterogeneous uptake coefficients and unclear reaction mechanisms applied in the model (Li et al., 2019; Xue et al., 2016; He et al., 2014). For details, see *Line 283-297*.

*Line 267: nonlinear behaviour in winter*

**Response: Accepted.** For details, see *Line 305*.

*Section 3.3.1 & 3.3.3, I miss here the discussion on the involvement of $SO_2$ emission changes to $PM_{2.5}$ aerosols and thereby to $O_3$ and nitrate aerosol*

**Response: Accepted.** A series of simulations ($C_{S-N}$; see *Line 160-165* and *Table 1*) are supplemented to show the responses of nitrate and $PM_{2.5}$ pollution to the $SO_2$ emission controls. The $SO_2$ emission reduction, although effective in reducing sulfate and $PM_{2.5}$ (*Fig. 12* in the revised manuscript), is not successful in regulating the nitrate pollution due to the chemical competition in nitrate and sulfate formations (Geng et al., 2017; Wang et al., 2013). Changes in nitrate concentration are linearly associated with the $SO_2$ emission reductions, with the β values of 2.90% during summer-autumn and 1.35% during winter. For details, see *Sec. 3.3.4* and *Fig. 12*.

[Figure]

**Fig. 12**. Responses of the surface nitrate (upper panels) and $PM_{2.5}$ (bottom panels) concentrations to the emission reduction scenarios of $NO_x$, $SO_2$, $NH_3$ and $VOC_s$ during Period I (a, b) and Period II (c, d).

*Line 310: have minor changes*

**Response: Accepted.** This sentence is rephrased as "The β values for surface sulfate change by −0.74%~+1.16% in BTH and −1.54%~+0.17% in YRD under the −10~−80% $NO_x$ emission reduction scenarios". For details, see *Line 348-350*.

*Lines 312, 313,345-348: How significant are these correlations?*

**Response: Accepted.** The $R^2$ values are supplemented in Eq. 1-3. For details, see *Line 351-352* and *Line 368-371*.

*Line 320: do not forget acid replacement reactions (e.g. reactions on dust or sea-salt aerosol)*

**Response: Accepted.** This sentence is rephrased as "The formation of nitrate mainly involves the $NO_2+OH{\rightarrow}HNO_3$ gas-phase oxidation and the heterogeneous hydrolysis of $N_2O_5$ and other nitrogenous gases". For details, see *Line 358-360*.

*Lines 324-325: do not forget here the reduction of $NO+O_3$ titration reaction*

**Response: Accepted.** This sentence is rephrased as "…due to the reduction of $NO+O_3$ titration reaction and the greater VOC availability in the warm season**".** For details, see *Line 365-366*.

*Line 337: reactions*

**Response: Accepted.** For details, see *Line 381*.

*Line 391 move 'during summer-autumn' to the end of line 392*

**Response: Accepted.** For details, see *Line 484*.

*Lines 408-415 and 419-427 would fit better in introduction or discussion. Please move appropriately.*

**Response: Accepted.** Three series of additional simulations (see *Line 160-165* and *Table 1*; $C_{S-N}$, $C_{N-N}$ and $C_{V-N}$) are conducted to show the responses of nitrate and $PM_{2.5}$ pollution to the cooperative controls of $NO_x$, $SO_2$, $NH_3$ and $VOC_s$, respectively. These texts have been moved to *Sec. 3.3.4*. For details, see *Sec. 3.3.4*.

**References:**

[revised manuscript text omitted]